# Evaluation of the Quality of Recovery from General Anesthesia in Dogs with Two Different Low Doses of Dexmedetomidine

**DOI:** 10.3390/ani14091383

**Published:** 2024-05-05

**Authors:** Chiara Di Franco, Irene Nocera, Pierre Melanie, Angela Briganti

**Affiliations:** 1Department of Veterinary Sciences, University of Pisa, 56124 Pisa, Italy; pierre.melanie@unipi.it (P.M.); angela.briganti@unipi.it (A.B.); 2Institute of Health Sciences, Sant’Anna School of Advanced Studies, 56127 Pisa, Italy; irene.nocera@santannapisa.it

**Keywords:** dogs, anesthesia, dexmedetomidine, recovery, delirium

## Abstract

**Simple Summary:**

During the awakening phase from general anesthesia, phases of agitation, excitement, vocalization, or violent movements are present in both human and veterinary medicine. Treatment for this phase involves the administration of sedatives such as alpha-2 agonist drugs that represents an established practice in equine anesthesia recovery. Dexmedetomidine is an alpha-2 agonist with sedative properties commonly used in veterinary clinical practice as a preanesthetic agent. The possibility of preventing these episodes of agitated awakening with the preventive administration of dexmedetomidine in canine patients has been studied in different populations. The objective of our study was to evaluate the quality of recovery from general anesthesia in dogs with the administration of two low doses of intravenous dexmedetomidine.

**Abstract:**

The purpose of this study was to evaluate the quality of recovery from general anesthesia with the administration of two low doses of dexmedetomidine in canine patients. For this blind randomized clinical trial study, 30 dogs undergoing general anesthesia for diagnostic procedures or elective surgery (ovariectomy/castration) were included. The patients were randomly divided into three groups, and at the end of anesthesia, they received a bolus of dexmedetomidine at 1 mcg/kg IV (D1), or a bolus of dexmedetomidine at 0.5 mcg/kg (D0.5), or a bolus of NaCl, in a total of 0.5 mL of solution for all three groups. After administration of the bolus, the anesthetist monitored the patients every 5 min by measuring heart rate, systolic and mean blood pressure, respiratory rate, and oxygen saturation. The quality of recovery was also assessed using 4 different scales. The extubation time, time of headlift, and standing position were also recorded. Both groups receiving dexmedetomidine had better awakening and a lower incidence of delirium when compared to saline administration. The heart rate was lower, while the systolic pressure was higher in the two groups D1 and D0.5 compared to the NaCl with a low presence of atrioventricular blocks. The extubation time resulted significantly higher in the D1 (17 ± 6 min) compared to the D0.5 (10 ± 4 min) and NaCl (8 ± 3 min) (*p* < 0.0001); the headlift time D1 (25 ± 10 min) resulted significantly longer than the NaCl group (11 ± 5 min) (*p* = 0.0023) but not than the D0.5 (18 ± 9 min). No significant differences were found among the three groups for standing positioning (D1 50 ± 18 min, D0.5 39 ± 22 min, NaCl 28 ± 17 min). The preventive administration of a bolus of dexmedetomidine at a dosage of 0.5 mcg/kg or 1 mcg/kg IV during the recovery phase improves the quality of recovery in patients undergoing general anesthesia.

## 1. Introduction

In pediatric patients, as in veterinary medicine, the management of recovery is a delicate phase during which episodes of agitation may occur [1,2,3,4]. The term dysphoria in the veterinary medical field is associated with behavior attributable to vocalizations, a lack of awareness of what is around, and restlessness [1], even if it does not currently have a unique recognized definition.

The incidence of dysphoria in veterinary literature varies from 22 to 35% and, although the mechanism of action is not entirely clear, the studies conducted highlight a high incidence during opioid administration [1,5,6]. Becker et al. describe the prevalence of dysphoria after intraoperative administration of fentanyl infusion in dogs undergoing stifle surgery. They reported the possible factors that influence dysphoria as follows: type of opioid or dosage used, individual dog susceptibility, or breed difference [1]. As already mentioned, the term dysphoria in veterinary literature is associated with opioids administration [1,3]. Numerous studies have been conducted in veterinary medicine in order to manage the awakening phase, especially in horses for which agitated recovery represents a higher problem to manage than in other species [7,8,9]. Agitation, excitement, vocalizations, or violent movements can, however, represent a danger not only for the animals, but also for the operator who follows the awakening phase, therefore, a correct management is essential for everyone’s safety. Dexmedetomidine, an alpha-2 agonist drug, has recently been studied in the recovery phase of equine, canine, and human patients [10,11,12,13].

Jarosinski et al. reported that the preventive use of a dosage of 0.5 mcg/kg of dexmedetomidine intravenously in dogs undergoing orthopedic surgery improved recovery quality, but prolonged recovery duration in sevoflurane-anesthetized healthy dogs [12]. The effects of dexmedetomidine on the cardiovascular system are well known, increasing peripheral vascular resistance with a reflex bradycardia, which can sometimes also cause atrioventricular blocks of varying severity [14].

Our hypothesis is that the use of prophylactic dexmedetomidine may be useful in allowing a better quality of recovery in patients not receiving intraoperative opioid infusions and in those undergoing diagnostic procedures. 

The objective of this study was to evaluate the effect of intravenous administration of two different doses of dexmedetomidine (0.5 mcg/kg and 1 mcg/kg) on the quality of recovery in patients undergoing general anesthesia. The secondary objective was to evaluate non-invasive hemodynamic parameters in these patients.

## 2. Materials and Methods

This prospective randomized blinded study was conducted at the Veterinary Teaching Hospital of the University of Pisa. This study received approval by the ethical committee for animal welfare of University of Pisa (n. 50/2023).

Canine patients undergoing general anesthesia for diagnostic procedures and elective surgeries (ovariectomy or orchiectomy) were included in the study; the owners signed a consent form to enroll their animal in the study.

All patients received a preanesthetic clinical examination and completed blood chemistry tests. Patients who required elective surgeries or diagnostic procedures, classified as ASA I, II, III, in which anesthesia lasted a minimum of 1 h, were included. The exclusion criteria were as follows: aggressiveness, neurological problems, heart disease under therapy, or the intraoperative administration of opioids infusion and/or rescue. Upon arrival, a behavioral category was calculated for each patient [15] and preanesthetic values (PRE), such as heart rate, systolic, mean, and diastolic blood pressure, respiratory rate, were registered. 

All patients were premedicated with the same anesthetic protocol: dexmedetomidine 5 mcg/kg (Dextroquillan 0.5 mg/mL, Fatro, Italy), methadone 0.2 mg/kg (Semfortan, 10 mg/mL; Dechra, Turin, Italy), and ketamine 1 mg/kg intramuscularly. Dogs were then induced with propofol (Proposure, Boehringer Ingelheim Animal Health, Noventana, Italy S.p.A.) to effect until endotracheal intubation was achievable and was maintained with isoflurane (Vetflurane, Virbac S.r.l., Milan, Italy). In case of surgeries (which included ovariectomy or orchiectomy), all patients received locoregional anesthesia with a bilateral quadratus lumborum nerve block (0.3 mL/kg for each point, with ropivacaine 0.5%) [16] or intratesticular (0.2 mL/kg for each testicle, with ropivacaine 0.5%) [17]. During the surgical/diagnostic procedure, patient monitoring was performed as routine. The heart rate, systolic, mean and diastolic blood pressure, oxygen saturation, respiratory rate, temperature, and end tidal CO_2_, (EtCO_2_) (Avance CS^2^ Pro, GE, Milan, Italy) were recorded every 5 min until the end of the procedure, when the administration of isoflurane was ceased. Patients were randomly divided (https://www.random.org) into three groups to receive one of the following treatments: dexmedetomidine 0.5 mcg/kg (D0.5 group), dexmedetomidine 1 mcg/kg (D1 group) or saline; all the treatments were diluted with NaCl to administer a total volume of 0.5 mL and were administered intravenously in 3 s by an anesthetist blind to the treatment. The patient was then disconnected from the anesthesia machine (Avance CS^2^ Pro, GE, Milan, Italy) and was taken to a dedicated room for recovery. Parameters, such as heart rate (HR), partial oxygen saturation (SpO_2_) (EDAN IM50, Sadel Medica, Pescara, Italy), respiratory rate (RR), systolic (SAP), and mean (MAP) arterial blood pressure with a non-invasive blood pressure monitor (PetTrust, BioCare, Aster Electrical Co. Ltd., Taoyuan City, Taiwan), were recorded at the end of the anesthetic procedure (T0), 5 min after the execution of the bolus (T1) and subsequently at 10, 15, 20, 25, 30 min until the patient was standing. During the recovery phase, all patients were positioned in sternal recumbency. 

In case of recovery characterized by agitation, excitement, vocalizations, or violent movements, the anesthetist administered a bolus of propofol 1 mg/kg and if the event was not resolved, another bolus of propofol was allowed.

The time to extubation, headlift, and positioning in standing were also recorded. Extubation was achieved when a prominent and persistent gag or swallowing reflex or chewing was noted by the blinded anesthetist. The degree of quality of awakening was recorded using different scales: visual analogic scale (VAS) [18], numerical rating scale (NRS) [18], simple descriptive scale (SDS) [10], and a modified version (SDS1) [18]. Each score scale was performed at the same moment and by the same clinician in a pre-determined random order. The degree of quality of awakening was evaluated in two moments: the early phase, which was immediately after the patient was extubated, and the late phase, when the patient was able to walk.

### Statistics

To detect a difference of 53% in the SDS recovery score between the groups, considering a maximum value of 3 as excellent recovery [10] with an α error of 0.05 the minimum number of dogs necessary for each group was 10, calculated using online software (Clincal.com). Data were analyzed for the distribution with a D’Agostino and Pearson. One way ANOVA with a Tukey post hoc or a Kruskal–Wallis with a Dunn’s post hoc test were used to compare the measurements for the three groups. One way ANOVA for repeated measures with a Dunnett’s post hoc test were used to compare the values recorded at T0 and all the subsequent monitoring times within each group. Parametrical data are presented ad mean and standard deviation and non-parametrical with median and range. Values of *p* < 0.05 were considered significant.

## 3. Results

A total of 30 patients were included, 10 per group as expected from the evaluation of the sample size. All patients completed the study uneventfully. There were no statistically significant differences between weight, age, pre-anesthesia behavioral category, the duration of anesthesia, isoflurane vaporizer setting at the end of anesthesia (Table 1), and the kind of procedure (diagnostic or surgical) between the three groups. The breeds included are as follows: Labrador Retriever (n = 11); Mixed Breed (n = 8); Bracco (n = 2); Caucasian Shepherd (n = 2); Lagotto Romagnolo (n = 1); Golden Retriever (n = 7); Beagle (n = 1); Jack Russel (n = 1); Australian Shepherd (n = 1); and Pointer (n = 1). The heart rate significantly decreased at T0 in comparison to pre-anesthesia in the three groups. At T5 and T10, the HR of the D1 and D0.5 groups was significantly lower than the NaCl group. Furthermore, in the D1 group, the HR was lower at all times compared to T0, while D0.5 was lower than T0 only for T10 and T15 (Figure 1).

Regarding the MAP, statistically significant differences were found between the MAP pre-anesthesia and T0 for all three groups. At T10 and T15, the MAP of Group D0.5 was significantly greater than NaCl group. At T5, T10 and T20 of the MAP of Group D1 was greater than T0, while in Group D0.5 the MAP values were higher at T15 and T20 compared to T0 (Figure 2).

Statistically significant differences were found between SAP pre-anesthesia versus T0 for D0.5 and NaCl. At T5, T10, T15, and T20, the SAP of Group D1 was greater than at T0, while in Group D0.5, it was only at T15 and T20 compared to T0. In the NaCl group, the MAP was different compared to T0 only at time T10 (Figure 3).

The respiratory rate was statistically significant different in all three groups between pre-anesthesia and T0 (Figure 4). The SpO_2_ was higher in the D1 group than in the NaCl group at T10 and T15 (Figure 5).

Concerning the recovering scores, in the early phase, all the scales used show a better quality of awakening with a statistically significant difference for the two DEX groups compared NaCl group (Figure 6).

In the late phase, only the VAS scale did not identify a difference in the quality of awakening between the D1 group and the NaCl group, despite the score range being higher in the NaCl group (Figure 7).

The extubation time resulted significantly higher in the D1 (17 ± 6 min) compared to D0.5 (10 ± 4 min) and NaCl (8 ± 3 min) (*p* < 0.0001). Regarding the headlift time, D1 (25 ± 10 min) resulted significantly longer than the NaCl group (11 ± 5 min) (*p* = 0.0023) but not longer than D0.5 (18 ± 9 min). No significant differences were found among the three groups for the standing positioning of patient (Figure 8) (D1 50 ± 18 min, D0.5 39 ± 22 min, NaCl 28 ± 17 min).

Three patients in the NaCl group required propofol rescue during the recovery phase, while no propofol rescue was necessary for the dexmedetomidine groups. Two patients in group D1, one patient in group D0.5, and one patient in the NaCl group presented second degree atrioventricular blocks approximately 10 min after the treatment administration, which was resolved spontaneously in a maximum of 10 min.

## 4. Discussion

The administration of a low dose of dexmedetomidine (0.5 or 1 mcg/kg) shows a better recovery quality when compared to the administration of saline solution. 

In the dexmedetomidine groups, no dogs required a bolus of propofol during the recovery phase, indicating that the use of dexmedetomidine enables the prevention of bad recovery. The recovery time, similar to the extubation time and headlift, was greater in the groups that received dexmedetomidine, but this can be considered acceptable because the maximum difference was about 10 min. The results emerged from our study are in line with previous studies conducted by Hunt et al. in 2014 and Jarosinski et al. in 2021 [10,12]. 

If we compare our results to the study of Hunt et al., in which 62.5 mcg/m^2^ (corresponding to about 3 mcg/kg) of dexmedetomidine was used, the headlift time and the time to standing were shorter for both our groups (D1 and D0.5). This result was expected due to the lower dosage of dexmedetomidine used. However, the NaCl group of the Hunt’s study needed a longer time for the headlift and the time to standing position. Since the duration of anesthesia was longer in our study (87 ± 50 min for Hunt’s vs. about 140 min for D1 and D0.5), the difference in recovery times can be due a different anesthesia management, which is an aspect that should be further investigated. 

In the study of Jarosinski et al., the time between the discontinuation of sevoflurane and the moment of extubation was named the “recovery time” and it is comparable to our extubation time. In our study, the extubation time resulted shorter compared to Jarosinski in all the groups, even though we used isoflurane instead of sevoflurane. This difference, although not clinically remarkable, can be related to the difference in administration time of the dexmedetomidine between the two studies, ten minutes before the discontinuation of the sevoflurane in the Jarosinski study, and at the moment of the isoflurane discontinuation, in our study. The fact that with isoflurane we recorded a shorter recovery time in comparison to sevoflurane can be related to the fentanyl infusion used and the longer anesthetic duration time in the Jarosinski study.

To date, there are no validated scales for evaluating the awakening phase in dogs. Despite this, all the scales that we used showed the same trend in both the early and late recovery phases. Dexmedetomidine provided a gentler awakening with a practically zero episodes of episodes of agitation, excitement, vocalizations, or violent movements. The mechanisms behind these attitudes in veterinary medicine have not yet been clarified, although it is known that the administration of opioids and inhalational anesthetics can contribute to a dysphoric awakening [1,12].

The hemodynamic alterations presented were a decrease in heart rate and an increase in the mean and systolic blood pressure in patients receiving dexmedetomidine, with a low incidence of atrioventricular blocks. The fact that the mean and systolic arterial pressure in the D1 and D0.5 groups was higher than in the NaCl group highlights the vasoconstriction effect of dexmedetomidine, which is physiological in normovolemic patients. The events of agitation, excitement, vocalizations, or violent movements requiring administration of propofol rescue occurred only in three patients who received the saline solution. Propofol, when rapidly administered intravenously, can cause a decrease in respiratory rate, even leading to apnea or respiratory arrest. Vasodilation is another side effect of rapid propofol administration, leading to a decrease in systemic pressure and possible compensatory tachycardia [19,20]. To date, there are no veterinary studies comparing the use of dexmedetomidine with propofol in dogs recovering from general anesthesia, thus further studies are needed to clarify the hemodynamic impact of these drugs in this phase.

The continuous monitoring of patients after the administration of the bolus highlighted a low incidence of second-degree atrioventricular blocks (also manifested in patients in the NaCl group) which spontaneously resolved without pharmacological intervention. 

The presence of first- and second-degree atrioventricular blocks has been reported in canine patients undergoing soft tissue or orthopedic surgery with a continuous infusion of dexmedetomidine at a dosage of 1, 2 or 3 mcg/kg/h with dose-dependent incidence [21]. This is relevant, as it allows us to state that the administration of dexmedetomidine at the dosages used in this study is safe and has a low hemodynamic impact, as underlined by researchers in previous studies [10,12].

The respiratory rate was not statistically different between the three groups. However, in absolute value, the patients who received NaCl in the last phase of awakening had an increase in the respiratory rate. The influence of dexmedetomidine on respiratory rate has not been fully clarified. A recent review analyzes how some studies report a decrease in respiratory rate, while others report the absence of respiratory changes when dexmedetomidine was administered intramuscularly [22]. A more gradual and gentle awakening in the groups that received dexmedetomidine could, however, explain the absence of an increase in respiratory rate in D1 and D0.5 group.

Regarding the oxygen saturation values, in our study, following the administration of 1 mcg/kg of dexmedetomidine at T10 and T15, the SpO_2_ value resulted statistically higher if compared to the NaCl group. In previous studies, after the administration of the dexmedetomidine bolus, patients remained on oxygen for at least 15 min to avoid desaturation and no significant differences between dogs which received dexmedetomidine versus those which received saline were found [10,12]. The mechanisms by which dexmedetomidine might improve SpO_2_ in sedated canine patients are unclear. It is reported that dexmedetomidine in continuous infusion at a dosage of 1 mcg/kg/h improves oxygenation and decreases intrapulmonary shunt [23], but in that study dogs were ventilated and under general anesthesia. Other studies report no changes in intrapulmonary shunt in dogs on spontaneous ventilation with a continuous infusion of dexmedetomidine at a dosage between 0.5 mcg/kg/h and 1 mcg/kg/h [24,25]. A possible explanation could be the movement of blood from the periphery to the central compartment secondary to vasoconstriction caused by dexmedetomidine; furthermore, the patients included in our study were all healthy, so the possible beneficial effect of dexmedetomidine increasing SpO_2_ did not lead to obvious clinical findings. The pulse oximeter probe was positioned on the patients’ tongue, for this reason it is important to underline that the values obtained may have been influenced by movement, an increased respiratory rate, and tremors that may characterize the awakening phase. A higher number of patients and serial arterial blood gases are necessary to try to understand the mechanism behind the results found.

This study presents some limitations. A higher number of cases could have led to different results concerning the incidence of bradycardia. In fact, the calculation of the sample size was performed to evaluate the quality of awakening, the primary objective of the study. Secondly, the same clinician performed all the four recovery scales, and this could have produced a bias in the subsequent scores. To limit this problem, the score was performed in a random sequence but the video recording of the animals during recovery would have been more appropriate to allow an unbiased evaluation of the 4 scales. This solution was not possible in our setting. Finally, to date, there is no validated scale for the evaluation of recovery from general anesthesia in dogs.

## 5. Conclusions

The preventive administration of a bolus of dexmedetomidine at dosage of 0.5 mcg/kg or 1 mcg/kg IV allowed us to correctly manage the recovery phase in canine patients who have not received intraoperative opioids infusions and in those undergoing diagnostic procedures, without creating important cardiac alterations. Dexmedetomidine administration induced longer extubation and headlift time, but this is not considered clinically relevant by the authors.

Further studies with a larger and more diverse population could be useful for a more conscious and free-use of this molecule.

## Figures and Tables

**Figure 1 animals-14-01383-f001:**
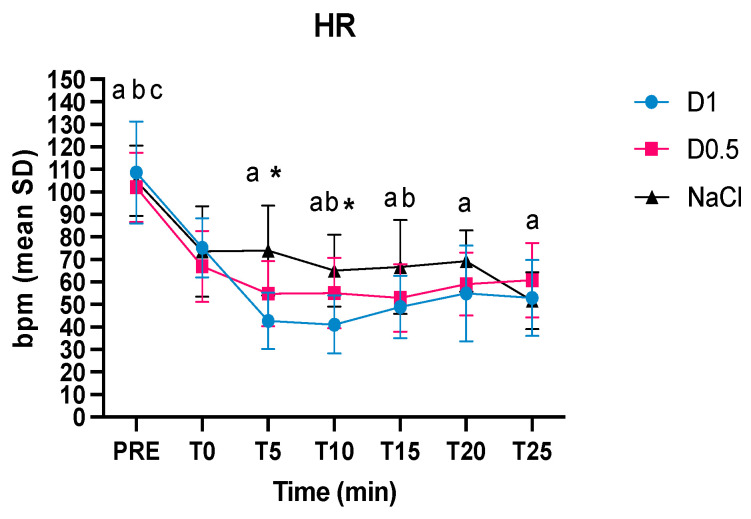
Intraoperative mean values and the standard deviations of the heart rate (HR) in the three groups, dexmedetomidine at 1 mcg/kg (D1), dexmedetomidine at 0.5 mcg/kg (D0.5), and normal saline (NaCl) before anesthesia, at the time of the bolus (T0), 5, 10, 15, 20, and 25 min after the administration of the bolus (T5, T10, T15, T20, and T25, respectively); * significantly different D1 from NaCl, ^a^ significantly different from T0 for group D1, ^b^ significantly different from T0 for group D0.5, ^c^ significantly different from T0 for group NaCl.

**Figure 2 animals-14-01383-f002:**
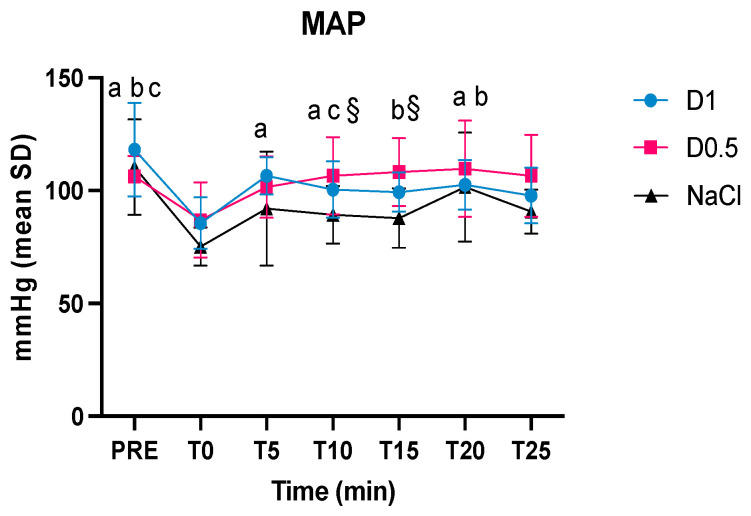
Intraoperative mean values and the standard deviations of the mean arterial pressure (MAP) in the three groups, dexmedetomidine at 1 mcg/kg (D1), dexmedetomidine at 0.5 mcg/kg (D0.5), and normal saline (NaCl) before anesthesia, at the time of the bolus (T0), 5, 10, 15, 20, and 25 min, after the administration of the bolus (T5, T10, T15, T20, and T25, respectively); § significantly different D0.5 from NaCl, ^a^ significantly different from T0 for group D1, ^b^ significantly different from T0 for group D0.5, ^c^ significantly different from T0 for group NaCl.

**Figure 3 animals-14-01383-f003:**
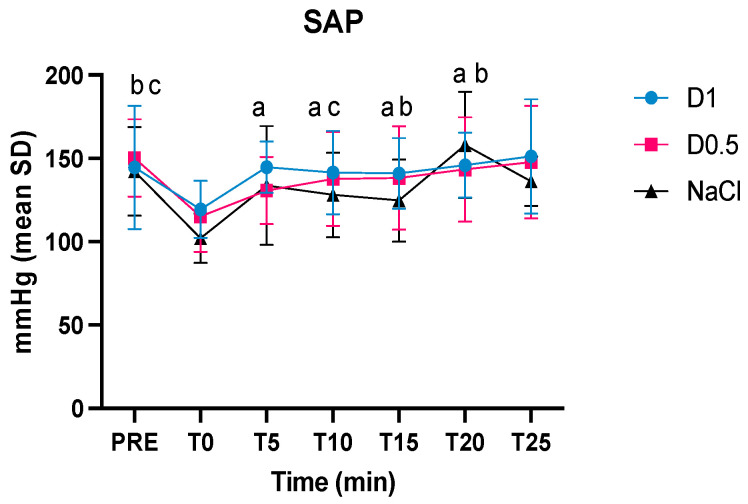
Intraoperative mean values and the standard deviations of the systolic arterial pressure (SAP) in the three groups, dexmedetomidine at 1 mcg/kg (D1), dexmedetomidine at 0.5 mcg/kg (D0.5), and normal saline (NaCl) before anesthesia, at the time of the bolus (T0), 5, 10, 15, 20, and 25 min after the administration of the bolus (T5, T10, T15, T20, and T25, respectively); ^a^ significantly different from T0 for group D1, ^b^ significantly different from T0 for group D0.5, ^c^ significantly different from T0 for group NaCl.

**Figure 4 animals-14-01383-f004:**
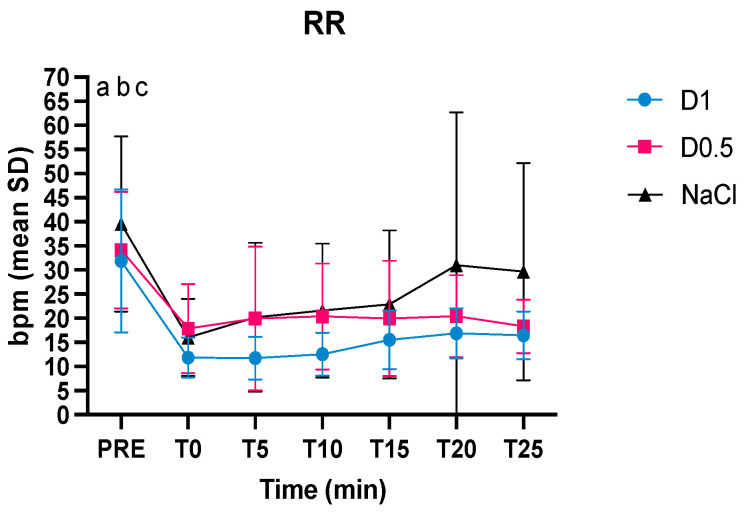
Intraoperative mean values and the standard deviations of the respiratory rate (RR) in the three groups, dexmedetomidine at 1 mcg/kg (D1), dexmedetomidine at 0.5 mcg/kg (D0.5), and normal saline (NaCl) before anesthesia, at the time of the bolus (T0), 5, 10, 15, 20, and 25 min after the administration of the bolus (T5, T10, T15, T20, and T25, respectively); ^a^ significantly different from T0 for group D1, ^b^ significantly different from T0 for group D0.5, ^c^ significantly different from T0 for group NaCl.

**Figure 5 animals-14-01383-f005:**
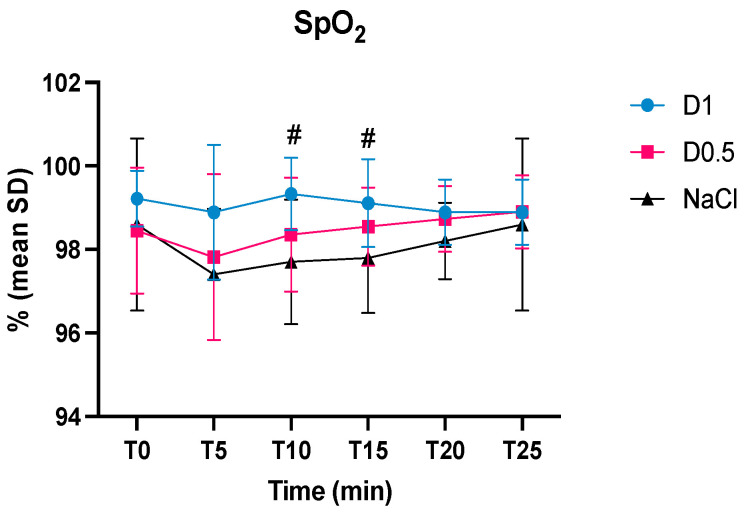
Intraoperative mean values and the standard deviations of oxygen saturation (SpO_2_) in the three groups, dexmedetomidine at 1 mcg/kg (D1), dexmedetomidine at 0.5 mcg/kg (D0.5), and normal saline (NaCl) before anesthesia, at the time of the bolus (T0), 5, 10, 15, 20, and 25 min after the administration of the bolus (T5, T10, T15, T20, and T25, respectively); ^#^ D1 different from NaCl.

**Figure 6 animals-14-01383-f006:**
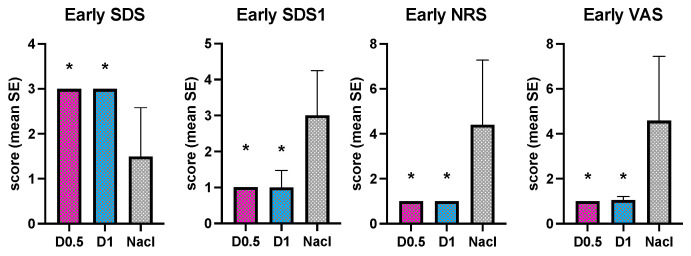
Intraoperative mean values and the standard deviations of Early SDS, SDS1, NRS and VAS in the three groups, dexmedetomidine at 1 mcg/kg (D1), dexmedetomidine at 0.5 mcg/kg (D0.5), and normal saline (NaCl); * different from NaCl.

**Figure 7 animals-14-01383-f007:**
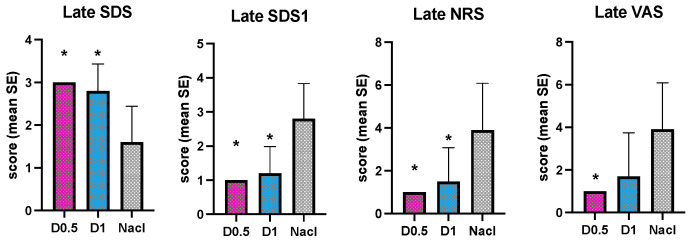
Intraoperative mean values and the standard deviations of late SDS, SDS1, NRS, and VAS in the three groups, dexmedetomidine at 1 mcg/kg (D1), dexmedetomidine at 0.5 mcg/kg (D0.5), and normal saline (NaCl); * different from NaCl.

**Figure 8 animals-14-01383-f008:**
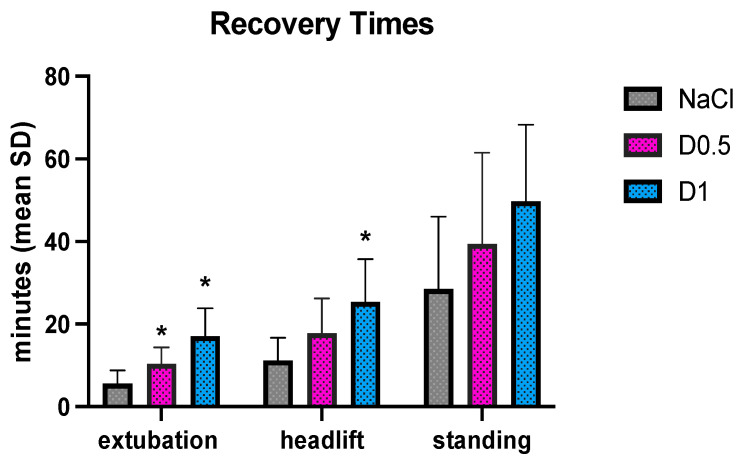
Mean values and the standard deviations of extubation, headlift, and standing in the three groups, dexmedetomidine at 1 mcg/kg (D1), dexmedetomidine at 0.5 mcg/kg (D0.5), and normal saline (NaCl); * different from NaCl.

**Table 1 animals-14-01383-t001:** Preoperative values and procedural findings of the three groups.

	D1	D0.5	NaCl
Weight (kg)	27 ± 6	26 ± 11	27 ± 9
Age (years)	7 ± 5	7 ± 4	5 ± 4
Pre-anesthesia behavioral category	2 (1–3)	2 (1–3)	2 (1–3)
Duration of anesthesia (min)	140 ± 50	142 ± 35	111 ± 37
Isoflurane vaporizer (%)	1.5 (1–1.5)	1.4 (1.1–1.5)	1.6 (1.1–1.8)
Surgery (%)	50	60	40
Diagnostic procedure (%)	50	40	60

## Data Availability

Data supporting the results stated above can be sent to anyone requesting them from the authors.

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
