# Peer review of "Evaluation of the Quality of Recovery from General Anesthesia in Dogs with Two Different Low Doses of Dexmedetomidine"

_animals, 2024, doi:10.3390/ani14091383_

Round 1

Reviewer 1 Report

Comments and Suggestions for Authors

Dear authors, I send you some comments I hope they help you to improve the manuscript. In general, the paper describes methods and results very well, needing improvement in the discussion. 

Author Response

Thank you for taking the time to review our paper.

Introduction 
Lines 56-57: In my opinion, I’ll change the second objective. You evaluated the following parameters: HR, SpO2 and non-invasive arterial pressure, so it was not an invasive hemodynamic evaluation (invasives arterial pressures, cardiac output, etc). For example: Secondary objective was to evaluate the non-invasive hemodynamic parameters. 
Thank you we corrected in the text
Material and Methods 
Lines 96-97: What system was used to record blood pressures?
We used a noninvasive blood pressure monitor (PetTrust, BioCare, Aster Electrical Co. Ltd., Taiwan) We have specified in the text as requested, thank you for the observation.
Results
Table 2: I think that Surgery and Diagnostic procedure is n, not % (in brackets). 
Corrected, thank you.
Discussion 
Lines 272-281: If the study of Jarosinski et al 2021 administered dexmedetomidine before than you did in your study (ten minutes before the discontinuation of sevoflurane), it doesn’t explain why they obtained longer extubation time. It would even be the other way around, that they would have recorded a shorter extubation time. I think, the main reasons were fentanyl infusion during anaesthesia procedure and the longer duration of anaesthesia. So, I think, you re-write this paragraph. 
We concluded exactly the same thing, and we reported it in the text.

Lines 282- 291: So, in the study of Hunts, the head lift and the time to standing were longer, due to the higher dexmedetomine dose used. However, if NaCl group (Hunts) was compared with Dexmedetomidine group (your study), you obtained shorter time. You mention that it could be due to the higher end-tidal isoflurane concentration used, but Hunts obtained a 1.6 % of isoflurane in S group and you used 1.6, too. You have to re- write this paragraph. 
Thank you for the observation, we have modified this paragraph.

Lines 301-303: The fact that the MAP and SAP in dexmedetomidine groups was higher than in the NaCl group highlight a more rapid recovery of vascular tone... of due to the vasoconstriction induced by alpha-two agonists.  Corrected

Lines 328-330: In your study, with D1, SpO2 was significantly higher than NaCL group in T10 and T15. But none of the animals in NaCL group presented hypoxemia, with SpO2 higher than 96%. So, in my opinion it is not a clinically significant difference. 
Thanks for the comment, probably it is not clinically relevant because the patients included in our study were all healthy patients. The effect of dexmedetomidine on oxygen saturation is not well studied yet, so this information could be interesting and to be evaluated for more studies regarding this aspect in different population.

Lines 341-342: I don`t understand the first limitation.
We changed the sentence.

Conclusion
What dose would you use, 0.5 or 1? With your results, could you choose one of them? 
It does not emerge from our study which of the two dosages is the best, which is why we have not included it in the conclusions.

Reviewer 2 Report

Comments and Suggestions for Authors

In this manuscript, the authors evaluate the effect of two doses of dexmedetomidine administered intravenously to control pain during the awakening phase from general anaesthesia.

A series of anaesthetic indices were evaluated, such as the evolution of cardiorespiratory activity and some clinical indices for evaluating anaesthetic efficiency (extubation time, head raising times standing position).

1. In the introduction, reference is made to the high incidence of dysphoria after the administration of opioids, the subject is insufficiently documented and I recommend the authors to develop this subject.

The introduction chapter should be reanalysed and developed.

2. The material and method chapter is difficult to understand, maybe a scheme illustrating the anaesthetic protocol used would be much better.

2. Regarding the material and method, the statistical analysis was carried out by comparison with the group that was administered saline solution, why was not a group introduced to which opioids were used in the anaesthetic protocol?

Author Response

Thank you for taking the time to review our paper.

1. In the introduction, reference is made to the high incidence of dysphoria after the administration of opioids, the subject is insufficiently documented and I recommend the authors to develop this subject. 
The introduction chapter should be reanalysed and developed.
We have expanded the introduction as requested.

2. The material and method chapter is difficult to understand, maybe a scheme illustrating the anaesthetic protocol used would be much better. 
We tried to ameliorate the paragraph.

2. Regarding the material and method, the statistical analysis was carried out by comparison with the group that was administered saline solution, why was not a group introduced to which opioids were used in the anaesthetic protocol?
Because it has already been studied by Jarosinski and colleagues in 2021, and also was not one of the objectives of our study. 

Reviewer 3 Report

Comments and Suggestions for Authors

Dear authors, I hope my comments will help you to improve the current version of the manuscript.

General comment

The topic studied by the authors is of relevance. However, the terms “delirium” and “dysphoria” have been used in an inappropriate way in the present manuscript. Indeed, their definition and differentiation are also unclear in humans, and almost nothing has been produced so far in the veterinary domain. What the authors have looked at is “quality of recovery”, terminology that I would suggest using along the whole text. The authors have not tried to develop a “delirium score”, but have just applied 4 (similar) scales to evaluate the recovery phase. The reason to do it is not clear, and no discussion has been provided.

Further focused studies are needed before trying to apply the terms “delirium” and “dysphoria” in animals.

Based on the comment above, I would also restructure all the paragraphs in which a mix between dysphoria/delirium has been done. Moreover, I would clearly differentiate the sentences and references that refer to humans and animals.

Please, carefully revise the punctuation used in the text.

Abstract

Line 24: what do you mean with “average systolic”?

Neither statistics nor numerical results are reported in the abstract. Currently, it looks more like a “simple summary”.

Introduction

As said, I would not speak about “delirium” or “dysphoria”, but rather of "quality of recovery". In the discussion session, I would then introduce the two terms and provide a discussion. Moreover, please clearly separate the literature on humans and veterinary species. Furthermore, differentiate when you speak about horses and small animals. It is important not to mix the different species, also to avoid future misleading referencing.

Having this in mind, please restructure the introduction.

Line 58 This hypothesis is strange. What do you mean with “useful”? Why would you expect dexmedetomidine to have an effect only in patients receiving intraoperative opioids? Moreover, you administer methadone to all the patients included in the study. Please rephrase.

M&M

Line 87 What do you mean with “as routine”? Please detail.

Line 89 The animals were randomized to receive a certain treatment, not “according to the treatment received”; right? Was the solution given IV?

How was the blood pressure measured? What was the position of the animal during the measurements?

Line 101 What did happen in case propofol was administered? Were the data from that patient excluded from the study, from that time point on?

Line 103 Do you mean “time to” the single event?

Line 109 Was the random order pre-determined?

Line 110 Was the early phase immediately after extubation, or when the animal raised the head?

What was the need to perform 4 similar scales? Which information do you expect to get from the single scales?

Line 127 Which program did you use to calculate the sample size? What is the meaning of the reference n.10 here? Please clarify.

Line 133 The is a cut sentence.

Results

Line 142-145 No statistic for these results has been reported in the M&M section. Please check.

Line 148 The possible fentanyl administration has not been mentioned before.

Line 149 The inclusion in the statistic of the “pre-anesthetic” values has not been previously reported in the text. The same for the following parameters reported in the study.

Table 2 I believe there is a mistake in “surgery” and “diagnostic procedure”. They are no %; am I right? Moreover, what do you mean by “mean summary” in the title?

Line 160 Do you mean D1?

The legend of the caption of Figure 5 is a bit unclear: “a” means that all the values with an “a” are different from the value of the group D1 at T0? This means, for example, that at T15 all the groups had a significant difference from the value of D1 at T0? The same for the following parameters reported in the text.

Line 213 Do you mean “of the two DEX”?

Line 258 When did the AV blocks occur?

Discussion

Line 296-297 Here references are needed. As mentioned before, it is important to separate the different species.

Line 301 What do you mean by “rapid recovery of the vascular tone”? It is known that the alpha-2 agonists induce vasoconstriction.

Line 306-314 This part can be cancelled for me. This comparison is out of topic here. Moreover, it would require a deeper look into the literature.

Line 315-320 Strange sentence formulation. Please rephrase. The core of the paragraph is that no strong cardiovascular side effects occurred at the dosage used; right? The discussion of this topic is currently a bit poor.

Line 321-324 Poor discussion. No reference to the possible influence of dex on the respiratory parameters?

Line 325-340 A long discussion for a not-so-relevant topic, looking at your results. Moreover, you do not discuss possible limitations in the SpO2 reading that could have brought to these results. I would suggest reducing this paragraph.

The paper was supposed to be focused on dysphoria/delirium, but a short part of the discussion is dedicated to that. For the way the manuscript is written, the reader has the feeling that the study has no main aim. Moreover, the discussion of the results remains a bit superficial and not exhaustive. I would suggest the authors to restructure the text of the manuscript, thinking about what was really important for them while performing the study (primary and secondary aims).

Author Response

Thank you for your time and for the comments.

General comment

The topic studied by the authors is of relevance. However, the terms “delirium” and “dysphoria” have been used in an inappropriate way in the present manuscript. Indeed, their definition and differentiation are also unclear in humans, and almost nothing has been produced so far in the veterinary domain. What the authors have looked at is “quality of recovery”, terminology that I would suggest using along the whole text. The authors have not tried to develop a “delirium score”, but have just applied 4 (similar) scales to evaluate the recovery phase. The reason to do it is not clear, and no discussion has been provided. 

Further focused studies are needed before trying to apply the terms “delirium” and “dysphoria” in animals. 

Based on the comment above, I would also restructure all the paragraphs in which a mix between dysphoria/delirium has been done. Moreover, I would clearly differentiate the sentences and references that refer to humans and animals.

Thanks for the comment, analyzing veterinary medicine articles in which the term dysphoria is used, we have modified by replacing the word delirium with dysphoria as suggested and we have checked the references.

Please, carefully revise the punctuation used in the text.
Done, thank you.

Abstract

Line 24: what do you mean with “average systolic”?
thank you for the comment, we removed it.

Neither statistics nor numerical results are reported in the abstract. Currently, it looks more like a “simple summary”.
Adding statistical or numerical results in the abstract would result in exceeding the word limit, but we would be happy to add at least some numerical results if the editor would allow us to increase till to 300 words.

Introduction

As said, I would not speak about “delirium” or “dysphoria”, but rather of "quality of recovery". In the discussion session, I would then introduce the two terms and provide a discussion. Moreover, please clearly separate the literature on humans and veterinary species. Furthermore, differentiate when you speak about horses and small animals. It is important not to mix the different species, also to avoid future misleading referencing.

Having this in mind, please restructure the introduction.

Corrected In the text. 

Line 58 This hypothesis is strange. What do you mean with “useful”? Why would you expect dexmedetomidine to have an effect only in patients receiving intraoperative opioids? Moreover, you administer methadone to all the patients included in the study. Please rephrase.

Dear reviewer, we indeed expect that dexmedetomidine would be useful also in patients which receive low or normal doses of opioids such as methadone in premedication. The veterinary literature reports dysphoria events mainly related to the administration of intraoperative opioids such as fentanyl infusion. Our hypothesis is based on the clinical experience acquired over the years regarding  several occasions in which it has been necessary to administer alpha-two agonists during dysphoric awakening in patients who have not received intraoperative opioids.

M&M

Line 87 What do you mean with “as routine”? Please detail.
Added in the text.

Line 89 The animals were randomized to receive a certain treatment, not “according to the treatment received”; right? Was the solution given IV? Corrected in the text.

How was the blood pressure measured? What was the position of the animal during the measurements? We used a noninvasive blood pressure monitor and the patients were in sternal recumbency; we have specified in the text as requested, thank you for the observation.

Line 101 What did happen in case propofol was administered? Were the data from that patient excluded from the study, from that time point on?
No the patients that received propofol were not excluded from the study.

Line 103 Do you mean “time to” the single event?
Thank you we have corrected in the text

Line 109 Was the random order pre-determined?
Yes, we added in the text, thank you for the suggestion.

Line 110 Was the early phase immediately after extubation, or when the animal raised the head?
After extubation, we corrected in the text.

What was the need to perform 4 similar scales? Which information do you expect to get from the single scales?

Thank you for pointing this out; there was not a real need, the problem is that a validated scale is not yet available so we thought we could evaluate the response of the animals by different scales, we know this is not a strong and valid point in fact we discussed this problem in the limitations. We are aware that the best method to verify the objectiveness of a scale would have been to record the recovery and make different people do all the scales, but it was not possible. If the reviewer thinks that all the scales are redundant in the paper, we are willing to remove some of them.

Line 127 Which program did you use to calculate the sample size? What is the meaning of the reference n.10 here? Please clarify.

Thank you, we added the program we used. The reference is referred to the median value for good recovery, and we added this in the text.

Line 133 The is a cut sentence.
Thank you for pointing this out, we removed this sentence.

Results

Line 142-145 No statistic for these results has been reported in the M&M section. Please check.
Thank you we changed the sentence in the text.

Line 148 The possible fentanyl administration has not been mentioned before. Thank you , we added the intraoperative fentanyl request as an exclusion criteria and removed this sentence.

Line 149 The inclusion in the statistic of the “pre-anesthetic” values has not been previously reported in the text. The same for the following parameters reported in the study.
Thank you we changed the sentence in the text.

Table 2 I believe there is a mistake in “surgery” and “diagnostic procedure”. They are no %; am I right? Moreover, what do you mean by “mean summary” in the title?

We corrected, thank you

Line 160 Do you mean D1?
Yes, corrected in the texts.

The legend of the caption of Figure 5 is a bit unclear: “a” means that all the values with an “a” are different from the value of the group D1 at T0? This means, for example, that at T15 all the groups had a significant difference from the value of D1 at T0? The same for the following parameters reported in the text.

Thank you for pointing this out; we corrected the legend in the text. We hope now it is clear.

Line 213 Do you mean “of the two DEX”? 
thanks we corrected as indicated.

Line 258 When did the AV blocks occur?
Added in the texts as requested.

Discussion

Line 296-297 Here references are needed. As mentioned before, it is important to separate the different species.
Added in the texts.

Line 301 What do you mean by “rapid recovery of the vascular tone”? It is known that the alpha-2 agonists induce vasoconstriction.
Corrected, thank you.

Line 306-314 This part can be cancelled for me. This comparison is out of topic here. Moreover, it would require a deeper look into the literature.
Thanks we would prefer to maintain this part.

Line 315-320 Strange sentence formulation. Please rephrase. The core of the paragraph is that no strong cardiovascular side effects occurred at the dosage used; right? The discussion of this topic is currently a bit poor.
Expanded discussion as requested.

Line 321-324 Poor discussion. No reference to the possible influence of dex on the respiratory parameters? Expanded discussion as requested.

Line 325-340 A long discussion for a not-so-relevant topic, looking at your results. Moreover, you do not discuss possible limitations in the SpO2 reading that could have brought to these results. I would suggest reducing this paragraph.  Thanks we would prefer to maintain this part. 

The paper was supposed to be focused on dysphoria/delirium, but a short part of the discussion is dedicated to that. For the way the manuscript is written, the reader has the feeling that the study has no main aim. Moreover, the discussion of the results remains a bit superficial and not exhaustive. I would suggest the authors to restructure the text of the manuscript, thinking about what was really important for them while performing the study (primary and secondary aims).

We tried to improve the discussion section

Round 2

Reviewer 3 Report

Comments and Suggestions for Authors

Dear authors,

thank you for the time spent in improving the manuscript following my suggestions.

I believe the manuscript still needs revision before publication. Please, find below my comments.

-          Reading the title and then the abstract it is not clear which was your primary aim. Was it the “quality of recovery” or the evaluation of the “cardiorespiratory variables”? Please, either change the tile or the main text. Based on this, I would ask to add results in the abstract (only the main results). If the editor allows you to expand your abstract, you could also include secondary results. As said last time, this way there are 2 simple summaries; this does not follow the requirements of the journal.

-          Linked to the previous comment. As mentioned last time, it is not clear what is your main aim. Your title speaks about recovery quality, but in the results you dedicate space to this only at the real end. Please modify the structure of your manuscript based on your priority.

-          In the previous review I have asked the authors not to use the terms delirium or dysphoria. In the current version, the unique term dysphoria has been used. As explained last time, I believe this is incorrect. Here I write again my previous comment: “The topic studied by the authors is of relevance. However, the terms “delirium” and “dysphoria” have been used in an inappropriate way in the present manuscript. Indeed, their definition and differentiation are also unclear in humans, and almost nothing has been produced so far in the veterinary domain. What the authors have looked at is “quality of recovery”, terminology that I would suggest using along the whole text.” If you want to keep one of the two words, I would then keep “delirium”, and not dysphoria. You did not inject opioids intra-op: why should you assess dysphoria? One of the few points that characterize dysphoria is the use of opioids. I would please ask the authors to clarify this point, that I believe is fundamental. Indeed, we should try to avoid publishing “foggy” information, that could lead to wrong referencing in the future.

Important: of the 4 scales you have used, only the SDS1 and SDS could spot clear signs of delirium. Thus, either you keep the 4 scales, but then you speak about “quality of recovery”, or you keep the SDS and the SDS1, and then you use the term “delirium”. Use the same terminology along the text.

Important (2): for the same reasons, when you speak about other papers where “quality of recovery” has been assessed, you cannot speak about delirium or dysphoria. Please, correct it along the text.

-          Current hypothesis: “Our hypothesis is that the use of prophylactic dexmedetomidine may be useful in reducing the incidence of dysphoria and allowing a better quality of recovery also in patients not receiving intraoperative opioid infusions and in those undergoing diagnostic procedures.”

I would suggest rephrasing as follows: “Our hypothesis is that the use of prophylactic dexmedetomidine may be useful in allowing a better quality of recovery (or in reducing the incidence of delirium, see above) in patients not receiving intraoperative opioid infusions and in those undergoing diagnostic procedures”. Still, it is a bit strange that in the previous version the fentanyl administration was foreseen (reported in the results), while in the current version it is an exclusion criteria. If you decide to test the recovery in patients not receiving opioids (due to the experimental protocol), you usually do not foresee an opioid as first rescue drug.

-          Why were the patients receiving propofol not excluded from the study? Did you give them a maximum score? How were the data analyzed?

-          Line 82 “lasted”

-          Specification of surgeries (currently at line 116) should go at line 81

-          Line 117 “with bilateral quadratus lumborum nerve block (0.3 mL/kg for each point, with ropivacaine 0.5%)

-          Line 118 “intratesticular block (0.3 ml/kg for each testicle, with ropivacaine 0.5%)

-          Line 120 I believe you “recorded” those parameters every five minutes.

-          Line 122-126 goes after line 126-127, thus you first have to say that the anesthesia was over, and then that one of the three treatments was administered.

-          Line 130 cancel the “:”, as well as the “,” at line 133 after the brackets.

-          Line 139 either “standing” or “recumbency”

-          Line 141 cancel “blinded”, you already mentioned before.

-          Line 173 Clinical.com does not allow you to have median in the sample size calculation page (am I missing something?). Moreover, it requires a SD, and either you assess the mean difference or a % increase/decrease. Please provide more information regarding the sample size calculation. It should be repeatable.

-          Line 187 “as”

-          Line 188 “statistically significant”. A value could also only have a clinical significance, and vice-cersa.

-          Line 196 “…were present”.

-          Line 196 “are”. Use always the same verb tense.

-          Line 200 Same comment of last time: “The inclusion in the statistic of the “pre-anesthetic” values has not been previously reported in the text. The same for the following parameters reported in the study.” You do not say what is “baseline” for you. When did you take it?

-          Line 200 not “versus” but “and”

-          In general, when you present the results, please do not mix the “within” with the “among” groups comparison. First you start with the within and then go to the among (or viceversa).

-          Line 257 “significant”

-          Line 318 please give the time “after the treatment administration”, not “isoflurane stop”.

-          Line 349-355 “This result was expected, due to the lower dosage of dexmedetomidine used. However, the NaCl group of the Hunt’s study needed a longer time to headlift and to standing position. Since the duration of anesthesia was longer in our study (87 ± 50 minutes for Hunt’s vs about 140 minutes for D1 and D0.5), the difference in recovery times can be due a different anesthesia management, aspect that should be further investigated.”

-          Line 357 “named”. Moreover, change “similar” to “comparable”.

-          Line 358 shorted compared to Jarosinski? Cancel also the sentence in brackets: all groups included also the D1.

-          Lines 363-365“The fact that with isoflurane we recorded a shorter recovery time in comparison to sevoflurane can be related to the fentanyl infusion used and the longer anesthetic duration had in the Jarosinski study”

-          Line376-378 Strange discussion. The fact that the pressure increases does not mean that the perfusion of the tissues increases. Alpha-2 agonists are well known to reduce the cardiac output. You do not want to start a physiological discussion here: I would remove this sentence.

-          Line 379 Propofol appears out of the blue. Please, remind the reader with few words your results, then you say that there are no vet studies, then you mention that dex is probably better than propofol due to the possible lower side effects.

-          Line 463 As you did for the other variables, you should start with your results, and then comparing them with the literature.

-          Limitations: if you decide to speak about delirium ,then you should mention the lack of validated scales too.

-          Limitations: if you performed a sound sample size calculation, the first sentence should be modified. You should rather say that you did a certain sample size calculation for your primary outcome, but that the sample size could not be appropriate for the secondary outcomes of your study. Before changing this part, you should first clarify (as mentioned above) which were the primary and secondary outcomes of your study.

-          Line 494 “to” not “of”

-          Line 495 please cancel “even in patients who have not received intraoperative opioids and in those undergoing diagnostic procedures”. This is not part of your study. You did not compare with and without opioids.

-          Line 497-498  “Dexmedetomidine administration induced longer extubation and head raising time, but this is not considered clinically relevant by the authors”

Author Response

Dear reviewer, 
Thank you for your time, and your effort to ameliorate our manuscript. We hope we addressed all your comments. 

-          Reading the title and then the abstract it is not clear which was your primary aim. Was it the “quality of recovery” or the evaluation of the “cardiorespiratory variables”? Please, either change the tile or the main text. Based on this, I would ask to add results in the abstract (only the main results). If the editor allows you to expand your abstract, you could also include secondary results. As said last time, this way there are 2 simple summaries; this does not follow the requirements of the journal. Corrected in the text.

-          Linked to the previous comment. As mentioned last time, it is not clear what is your main aim. Your title speaks about recovery quality, but in the results you dedicate space to this only at the real end. Please modify the structure of your manuscript based on your priority. Done, thank you.

-          In the previous review I have asked the authors not to use the terms delirium or dysphoria. In the current version, the unique term dysphoria has been used. As explained last time, I believe this is incorrect. Here I write again my previous comment: “The topic studied by the authors is of relevance. However, the terms “delirium” and “dysphoria” have been used in an inappropriate way in the present manuscript. Indeed, their definition and differentiation are also unclear in humans, and almost nothing has been produced so far in the veterinary domain. What the authors have looked at is “quality of recovery”, terminology that I would suggest using along the whole text.” If you want to keep one of the two words, I would then keep “delirium”, and not dysphoria. You did not inject opioids intra-op: why should you assess dysphoria? One of the few points that characterize dysphoria is the use of opioids. I would please ask the authors to clarify this point, that I believe is fundamental. Indeed, we should try to avoid publishing “foggy” information, that could lead to wrong referencing in the future. 

Important: of the 4 scales you have used, only the SDS1 and SDS could spot clear signs of delirium. Thus, either you keep the 4 scales, but then you speak about “quality of recovery”, or you keep the SDS and the SDS1, and then you use the term “delirium”. Use the same terminology along the text.

Important (2): for the same reasons, when you speak about other papers where “quality of recovery” has been assessed, you cannot speak about delirium or dysphoria. Please, correct it along the text. Corrected in the text.

-          Current hypothesis: “Our hypothesis is that the use of prophylactic dexmedetomidine may be useful in reducing the incidence of dysphoria and allowing a better quality of recovery also in patients not receiving intraoperative opioid infusions and in those undergoing diagnostic procedures.”

I would suggest rephrasing as follows: “Our hypothesis is that the use of prophylactic dexmedetomidine may be useful in allowing a better quality of recovery (or in reducing the incidence of delirium, see above) in patients not receiving intraoperative opioid infusions and in those undergoing diagnostic procedures”. Still, it is a bit strange that in the previous version the fentanyl administration was foreseen (reported in the results), while in the current version it is an exclusion criteria. If you decide to test the recovery in patients not receiving opioids (due to the experimental protocol), you usually do not foresee an opioid as first rescue drug. 

Rephrase as suggested. Regarding the administration of fentanyl, the first version of the manuscript was inaccurate.

-          Why were the patients receiving propofol not excluded from the study? Did you give them a maximum score? How were the data analyzed? Patients who received propofol were those who received a high score on the early scale and subsequently obtained a lower score on the late scale. Thanks for your comment.

-          Line 82 “lasted” Corrected, thank you.

-          Specification of surgeries (currently at line 116) should go at line 81 Done.

-          Line 117 “with bilateral quadratus lumborum nerve block (0.3 mL/kg for each point, with ropivacaine 0.5%) Corrected, thank you.

-          Line 118 “intratesticular block (0.3 ml/kg for each testicle, with ropivacaine 0.5%) Corrected, thank you.

-          Line 120 I believe you “recorded” those parameters every five minutes. Corrected in the text.

-          Line 122-126 goes after line 126-127, thus you first have to say that the anesthesia was over, and then that one of the three treatments was administered. Done, thank you for the suggestion.

-          Line 130 cancel the “:”, as well as the “,” at line 133 after the brackets. Corrected, thank you.

-          Line 139 either “standing” or “recumbency” Corrected, thank you.

-          Line 141 cancel “blinded”, you already mentioned before. Done, thank you.

-          Line 173 Clinical.com does not allow you to have median in the sample size calculation page (am I missing something?). Moreover, it requires a SD, and either you assess the mean difference or a % increase/decrease. Please provide more information regarding the sample size calculation. It should be repeatable. Thank you for your comment, we corrected the text accordingly.

-          Line 187 “as” Done.

-          Line 188 “statistically significant”. A value could also only have a clinical significance, and vice-cersa. In the results section we just reported the statistical results. We commented the results in the discussion.

-          Line 196 “…were present”. We rephrase this sentence.  

-          Line 196 “are”. Use always the same verb tense. We rephrase this sentence.  

-          Line 200 Same comment of last time: “The inclusion in the statistic of the “pre-anesthetic” values has not been previously reported in the text. The same for the following parameters reported in the study.” You do not say what is “baseline” for you. When did you take it? We recorded preanesthetic values before anesthesia, at the time of behavioral category assignment; we included in the text, thanks for the comment.

-          Line 200 not “versus” but “and” Corrected.

-          In general, when you present the results, please do not mix the “within” with the “among” groups comparison. First you start with the within and then go to the among (or viceversa). Thank you, corrected

-          Line 257 “significant” Corrected.

-          Line 318 please give the time “after the treatment administration”, not “isoflurane stop”. Corrected, thank you.

-          Line 349-355 “This result was expected, due to the lower dosage of dexmedetomidine used. However, the NaCl group of the Hunt’s study needed a longer time to headlift and to standing position. Since the duration of anesthesia was longer in our study (87 ± 50 minutes for Hunt’s vs about 140 minutes for D1 and D0.5), the difference in recovery times can be due a different anesthesia management, aspect that should be further investigated.” Corrected, thank you.

-          Line 357 “named”. Moreover, change “similar” to “comparable”. Done.

-          Line 358 shorted compared to Jarosinski? Cancel also the sentence in brackets: all groups included also the D1. Done.

-          Lines 363-365“The fact that with isoflurane we recorded a shorter recovery time in comparison to sevoflurane can be related to the fentanyl infusion used and the longer anesthetic duration had in the Jarosinski study” Corrected, thank you.

-          Line376-378 Strange discussion. The fact that the pressure increases does not mean that the perfusion of the tissues increases. Alpha-2 agonists are well known to reduce the cardiac output. You do not want to start a physiological discussion here: I would remove this sentence.

Removed as requested.

-          Line 379 Propofol appears out of the blue. Please, remind the reader with few words your results, then you say that there are no vet studies, then you mention that dex is probably better than propofol due to the possible lower side effects. Thanks anyway for the suggestion on the structure of the discussion, we have modified the sentences.

-          Line 463 As you did for the other variables, you should start with your results, and then comparing them with the literature. Done, thank you.

-          Limitations: if you decide to speak about delirium ,then you should mention the lack of validated scales too. Corrected, thank you.

-          Limitations: if you performed a sound sample size calculation, the first sentence should be modified. You should rather say that you did a certain sample size calculation for your primary outcome, but that the sample size could not be appropriate for the secondary outcomes of your study. Before changing this part, you should first clarify (as mentioned above) which were the primary and secondary outcomes of your study. Corrected in the text, thank you.

-          Line 494 “to” not “of” Corrected, thank you.

-          Line 495 please cancel “even in patients who have not received intraoperative opioids and in those undergoing diagnostic procedures”. This is not part of your study. You did not compare with and without opioids. Corrected as requested.

-          Line 497-498  “Dexmedetomidine administration induced longer extubation and head raising time, but this is not considered clinically relevant by the authors” Corrected in the texts, thank you.